# B3010 : A Boosted TSI 3010 CPC for Airborne Studies

David PICARD[1], Michel ATTOUI[2], and Karine SELLEGRI[1]

[1]LaMP, CNRS, Université Clermont Auvergne
[2]LISA, CNRS, Université Paris Est à Créteil

**Correspondence:** David PICARD (d.picard@opgc.univ-bpclermont.fr)

**Abstract.** In the present paper, we expose how we boosted the performances of a commercial CPC model TSI 3010 to detect particles as small as 1.5 nm, while preserving the robustness and reliability of the original instrument. The TSI 3010 was selected for our deep knowledge of its internals, and its large incorporated butanol reservoir that allows continuous operation for several hours without refill, which is well suited to airborne operation. Besides, it is still pretty easy to buy instruments

5   from the TSI 3010 family from companies that specialize in used scientific instruments retail. The CPC described in this study is called *B3010* hereafter, where the "B" stands for *boosted*. We provide an evaluation of its performances down to 1 nm using standard calibration methods, and comparisons with ultrafine CPCs (TSI 3025 and TSI 3776) as well as with its original version. One important application of the B3010 being high-altitude measurement stations and airborne studies, the instrument's detection efficiency was quantified for various inlet flow rates and pressures.

## 1 Introduction

### 1.1 Scientific background

It is now widely admitted that atmospheric particles have an impact on climate and health. Their concentration in the atmosphere is largely determined by their sources, that can be primary (mecanically emitted), or secondary (from a gaz-to-particle

15   conversion process within the atmosphere). Among the formation pathways of secondary aerosols, nucleation is the process responsible for the formation of new nanoparticle clusters (as opposed to the process of condensation onto pre-existing particles). Triggered by photochemical processes, oxidized lower volatility products are formed, of which some have the properties of nucleating into new particulate clusters. Once particle clusters are formed by nucleation, they may be lost onto pre-existing particles by coagulation if they do not rapidly grow to larger sizes by condensation of less volatile, but more abundant species.

20   The processes of nucleation and early growth lead to the occurrence of New Particle Formation (NPF) in the atmosphere. NPF occurs over several hours, and is considered responsible for generation of a large number of aerosols at the global scale (Spracklen et al. (2006)). With the development of instruments detecting particles of nanometric size, NPF events have been observed in a growing number of environments (Kulmala et al. (2004)).

In particular, at high altitudes, NPF events have been recorded with a high frequency in the French central mountain chain (Venzac et al. (2007), Boulon et al. (2011)), in the Alps (Boulon et al. (2010)), in the Himalayas (Venzac et al. (2008)) and Bolivian Andes, at 5200 m a.s.l. where the frequency of NPF events are amongst the highest in the world (Rose et al. (2015)). It is important to characterize the mechanisms and gas-phase precursors to NPF specific to high altitudes, because particles have a longer life time in this part of the atmosphere, and because they are in direct interaction with the environmental conditions conducive to cloud formation. The chemical species responsible for the formation of embryos and the ones responsible for their growth to larger sizes may be different. Therefore, in order to understand nucleation processes, it is essential to detect the embryos of particles before they are lost or grown to larger sizes. It is estimated that the first embryos of stable particles in the atmosphere have a size of 1 nm.

Diurnal conditions are necessary for the study of nucleation because they determine the presence of photochemical processes at the origin of nanoparticle precursor gases. High altitude stations, however, are frequently influenced by the uplifted air masses during the day due to forced convection on mountainous slopes, or to natural heat convection. Therefore, it is relatively rare to meet the appropriate conditions for the study of the nucleation process taking place above the atmospheric boundary layer from the ground measurement stations. Airborne measurements offer a much higher potential, not only to overcome artefacts related to the topography of ground stations, but also to evaluate the spatial (horizontal and vertical) extension of the process and to reach specific aerosol plumes (for instance desert dust, or volcanic ash in which the nucleation process could be favored). In the past, instrumentation embedded in an aircraft has been able to detect newly formed particles in a size range between 5 and 10 nm (Rose et al. (2015)) showing that the frequency of occurrence of ultrafine particles was maximum in the 2000-3000 m altitude range. Both for mountain-top and aircraft-based measurements, there is a need to measure nanoparticle concentrations with a controlled inlet flow rate and well-characterized low-pressure performances.

## 1.2 Technical background

The TSI 3010 CPC is a later version of the TSI 3760, designed by P. Keady (Keady (1988)), and targeted at particle concentration monitoring in clean rooms, as can be found in the pharmaceutical and electronics industries. The TSI 3760 and other models of the same product line (TSI 3762 and TSI 3762A) are all based on the same compact and clean room compatible design. The saturator is a reservoir for the working fluid and allows several days of continuous operation. The condenser is cooled by a thermoelectric cooler (TEC), sandwiched between the condenser and a heat sink. The heat sink evacuates the heat from the hot side of the TEC and channels it to the saturator. The TEC thus cools the condenser and heats the saturator at the same time. The operating temperature $T_S$ of the saturator is typically a few degrees above the ambient temperature. In order to prevent contamination of the ambient air (foremost specification in clean rooms), the instrument has no moving parts (no fan nor pump). Unlike most other CPCs, the temperatures of the condenser $T_C$ and the saturator $T_S$ are not controlled independently. Instead, the temperature difference $\Delta T$ between the condenser and the saturator is maintained constant. The thermal design was particularly well-thought, and ensures that $T_S$ is within a adequate range to saturate the sample flow with butanol vapor under a wide range of operating conditions. The optical detector block is in thermal contact with the heat sink, which keeps it warm enough to prevent the butanol vapor to condense on the lenses.

The TSI 3760 controls the sample flow with a critical orifice, and needs a vacuum pump to operate. In order to further reduce the risk of contamination, a second critical orifice is used to flush the air from the inner volume of the CPC housing. This is called the purge flow. The slight under-pressure in the housing causes any particle to be evacuated to the vacuum pump. In addition, the purge flow helps cool the electronics.

Back in 1988, all butanol-fueled CPCs had a sample flow rate of 0.3 l/min, starting with the TSI 3020. TSI later introduced

the TSI 3022, featuring a 1.2 l/min bypass flow, and the same 0.3 l/min flow rate in the optical detector. The bypass flow, called "make-up air" reduces diffusion losses. Those 1.5 l/min CPCs were called "high flow", while the legacy models were called "low flow".

Keady's TSI 3760 doesn't have such a bypass. It operates with a sample flow rate of 1.415 l/min and a purge flow rate of 1.4 l/min, making up 2.8 l/min, about 1/10 ft$^3$/min.

The minimum size of the particles that can act as condensation nuclei depends on the supersaturation ratio of the vapor of the working fluid in the cooled condenser. The smaller the particle, the higher the supersaturation ratio required to initiate the vapor-to-droplet conversion (nucleation). The supersaturation profile in the condenser depends on the flow rate, the vapor-saturated air thermodynamic properties, and condenser temperature $T_C$. The supersaturation ratio peaks at a distance past the entrance of the condenser that depends on the parameters listed here-above. This is where particles activate. The remaining of

the condenser beyond the maximum supersaturation point is simply used to grow the droplets to a detectable size (*ca.* 1 μm).

The higher the flow rate, the farther the supersaturation peak from the entrance of the condenser. Keady's design (Keady (1988)) keeps this distance short without increasing the length of the condenser, by splitting the sample flow into 8 short tubes. The flow rate in each tube is 0.177 l/min, resulting in a total sample flow rate of 1.415 l/min. The upside of the multi-tube design is a very compact instrument. The downside, because of the small flow rate, is higher particle losses (diffusion for the

smaller and transport for the bigger). Finally, with this settings, the cut-off diameter at 50 % detection efficiency, noted $D_{P50}$, is 11 nm.

The optical detector features a 180° layout, where the laser diode faces the photo-detector. This design is tailored for clean rooms environments with ultra low particle concentrations. Nevertheless, it performs well up to concentrations of $10^4$ #/cm$^3$.

The optical detector was made to count single particles. When a particle crosses a laser beam, light is scattered, sensed by

the photo-detector, which in turn generates an electrical pulse. The pulse is conditioned, then captured by a digital counter. As the flow rate is constant, it is easy to calculate the particle number concentration. At high concentration, the probability for two particles or more to overlap as they cross the beam increases. Then, only one pulse is generated as several particles traverse the detector. This phenomenon, known as coincidence, results in under-counting. A correction method based on Poisson's equation (Pisani and Thomson (1971), Gebhart (2001)) is implemented by the following equation :

$$N_a = N_i \cdot e^{N_a \cdot Q \cdot t} \tag{1}$$

Where $N_a$ is the actual concentration (#/cm$^3$), $N_i$ is the indicated or measured concentration (#/cm$^3$), $Q$ is the flow rate (cm$^3$/s) and $t$ is the effective time each particle resides in the viewing volume. The $N_a$ in the exponent can be approximated by $N_i$. In the first TSI 3760, Q = 1.4 l/min and t = 0.25 μs, resulting in a coincidence error of only 6 % at $10^4$ #/cm$^3$.

Although the TSI 3760 was designed for the clean rooms market, the good performances and affordable price helped make it popular in a wide range of applications, including atmospheric research (Ström and Ohlsson (1998), Woo et al. (2001)). Following this success, TSI introduced the now well-known TSI 3010. This model boasts significant improvements, such as a tighter temperature control, and, most importantly, the ability to drive a differential mobility analyzer (DMA, Knutson and Whitby (1975)) in an SMPS system (Wang and Flagan (1990)). Indeed, the TSI 3010 embeds a DAC (digital-to-analog converted) whose voltage can be set by a command sent to the RS-232 serial port interface. The TSI 3010 is based on the same concept as the TSI 3760. It has a larger liquid reservoir. But the main change is the sample flow rate, 1.0 l/min, compared to 1.4 l/min in the TSI 3760. This affects the dimensions of the tubes in the condenser, in order to maintain the same cut-off diameter (D$_{P50}$) around 10 nm.

Thanks to its 10 nm cut-off diameter and low price compared to a ultrafine sheathed CPC, the TSI 3010 was soon widely adopted in SMPS systems, covering many fields, including laboratory experiments and field measurements (O'Dowd et al. (1998), Schröder and Ström (1997), O'Dowd et al. (2009), O'Dowd et al. (2007)). Equivalent models including TSI 3760A, TSI 3762 also became very popular for the same reasons. Bricard's work at the Puy de Dome station (1465 m a.s.l.), France, leveraging the first continuous flow CPC (Bricard et al. (1972), El Golli et al. (1975)), shed light on the need to measure aerosol number concentrations on mountain tops or in research aircrafts. The TSI 3010 did play a key role in this field too (Seifert et al. (2004)).

The TSI 3010 is marketed with a cut-off diameter of 10 nm, when operated at the default temperature gradient ΔT = T$_S$ - T$_C$ = 17°C. The user can change ΔT by issuing a command on the RS-232 interface. The detection efficiency can thus be improved easily by increasing ΔT, without modifying the CPC. However, ΔT is coerced to a *safe* range in the firmware, so as to protect the TEC, and prevent homogeneous nucleation of the butanol vapor to occur. The principle of operation of the CPC is based on heterogeneous nucleation, where vapor condenses on the particles, that play the role of condensation sites, or seeds. On the opposite, homogeneous nucleation forms droplets without a seed particle. This latter process is an unwanted side-effect that must be avoided. The onset of homogeneous nucleation imposes the maximum ΔT for a given flow rate in a given saturator — condenser geometry.

Mertes et al. (1995) measured the cut-off diameter for different temperature gradients ΔT varying from 17 to 25 °C down to 5 nm. We note that the cut-off diameter of the TSI 3025, a sheathed CPC, is 2.7 nm with a sample flow rate of only 0.3 l/min (Kesten et al. (1991)). With a custom-built TSI 3010-like CPC with a heating mat on the saturator, an additional TEC on the condenser and a modified EEPROM (non-volatile memory integrated circuit), Russell et al. (1996) were able to reach independently T$_S$ = 38 °C and and T$_C$ = 2 °C respectively (ΔT = 36 °C). They measured the detection efficiency with aerosols of salt (NaCl) and silver as small as 4.5 nm. No homogeneous nucleation was detected with a 1 l/min particle-free air stream at ΔT = 36 °C. During an inter-comparison study of the size dependent detection efficiency of 26 CPCs (Wiedensohler et al. (1997)) showed that for the TSI 3010 operating at ΔT = 36 °C and Q = 1 l/min, the cut-off diameter of silver particles

is 3.75 nm, without homogeneous nucleation. At last, it was found that for a given temperature gradient $\Delta T$, the detection efficiency is higher if the condenser temperature $T_C$ is lower (Barmpounis et al. (2018)).

## 2 Design of the B3010

CPCs can be separated in two main categories : non-sheathed sample flow (Bricard et al. (1976), Keady (1988)) and sheathed sample flow (Stolzenburg and McMurry (1991)). The former are more robust and cheaper, while the latter boast a higher
detection efficiency of sub-10 nm particles and a lower probability of coincidence at high particle concentrations. Indeed, as noticed in Wiedensohler et al. (1994), the very low aerosol flow rate in sheathed CPCs, such as the TSI 3025 may lead to systematic deviations and statistical uncertainties in low concentration situations. The instrument described here belongs to the first category of non-sheated CPCs.

The goal of this development, encouraged by a recent study by Kangasluoma et al. (2015), that demonstrated the possibility
to detect sub-3 nm particles by just changing the temperatures in a standard TSI 3772 (successor of the TSI 3010), is to approach the performances of a UCPC, but sticking to a simple and robust design.

As we target airborne measurements, aircraft safety rules and specific constraints made the design process somewhat more complex. Airborne requirements include the ban of external butanol fill bottles, the need for all instruments to connect to common inlet and exhaust lines in order to avoid a critical cabin pressure drop. In other words, the inner flow paths of the
instruments are at outside ambient pressure, while the rest of the instruments are at cabin pressure. Besides, the power supplies found in aircrafts, can produce large voltage transients, possibly causing permanent damage to electronic devices.

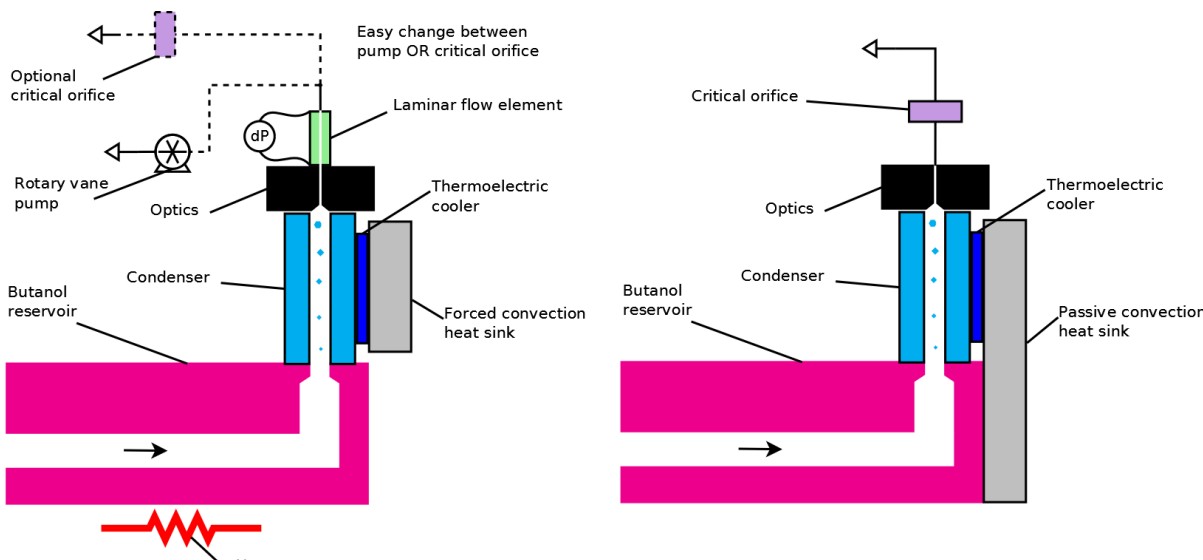

**Figure 1.** B3010 (left) and TSI 3010 (right) diagrams

In order to reduce the development time, our design reuses the saturator, condenser and optics of an original TSI 3010. Everything else was redesigned, involving 3D CAD modeling, electronics and software design. The saturator taken from the TSI 3010 is also a reservoir, that can hold more working fluid (butanol) than needed for a standard flight of 4 to 6 h, thus eliminating the need for an external fill bottle.

The key parameters that govern the cut-off diameter of a CPC are the volume flow rate, and the temperature gradient between the saturator and the condenser. In order to gain full control over the supersaturation process, it was necessary to control each of them separately. Indeed, the original TSI 3010 uses a thermoelectric cooler (TEC) to pump the heat out of the condenser and into the saturator, thus acting as a cooler and a heater at the same time. The temperature gradient is kept constant, but without control on the condenser absolute temperature. Besides, the flow rate is set by a critical orifice and is not measured. The flow rate cannot be changed unless the orifice is replaced.

For our purpose, we had to break the thermal bond between the hot side of the TEC and the saturator block. The tall heat sink was replaced by a smaller one, but with forced convection. The TEC was replaced by two TECs connected in series. Resistive heaters were stuck on the saturator block. In addition, in order to prevent the butanol to condense on the optics, a heater was added on the optical block. The optical block temperature is kept above *ca.* 40°C. Finally, we added the option to switch easily between the critical orifice and a small rotary vane pump to suit all use cases. The pump allows to adjust the flow rate at will, and removes the need for a bulky external vacuum pump.

We measure the flow rate with a laminar flow element, corrected for absolute pressure. The absolute pressure is measured by a miniature sensor, connected to the optical chamber with a capillary tube. The pressure intake is centrally located, between the saturator-condenser block and the laminar flow element. The volume flow rate is calculated from the differential pressure measured by a 50 Pa miniature sensor accross a laminar flow element and compensated for absolute pressure. The volume flow rate was calibrated with a Drycal Gilibrator bubble volume flow meter for a number of absolute pressures. The volume flow rate is a key measurement, since it is used to calculate the particle number concentration. The concentration C is calculated from the number of particle counts N, accumulated during $T_S$, at a volume flow rate $Q_V$, according to Eq. 2.

$$C = \frac{N}{Q_V \cdot T_S} \tag{2}$$

The electronic boards of the CPC were redesigned from scratch. The power supply board was designed to operate off aircrafts 28 VDC with special care taken to stand reverse polarity and load dumps. The power supply can stand $\pm 80$ V overvoltages, and features an overcurrent protection (current limiting). A second board is used to control the current in the TECs with fast, high power MOSFET transistors in a half-bridge. Finally a sensor board with an 8-bit microcontroller measures pressures, temperatures, flow rate, counts the pulses from the optics and generates 300 kHz waveforms to drive the TEC power supply MOSFETs.

The system is controlled by a credit card-sized computer board powered by an ARM processor. The operating system is a custom-made Linux system, built from scratch with the Buildroot framework. The computer runs advanced software algorithms (Landau (1993)) to achieve tight control of the flow rate (+/-0.02 l/min) and of the temperatures (+/-0.1°C). The B3010 features

a TSI compatible serial port command set, a BNC pulse output connector, data recording capability, and a 3.8" touchscreen. An Ethernet port allows remote access while the software is running, due to the multitasking nature of Linux. The software is written in C++, with the Qt open source library.

## 3    Calibration bench

### 3.1    Experimental setup

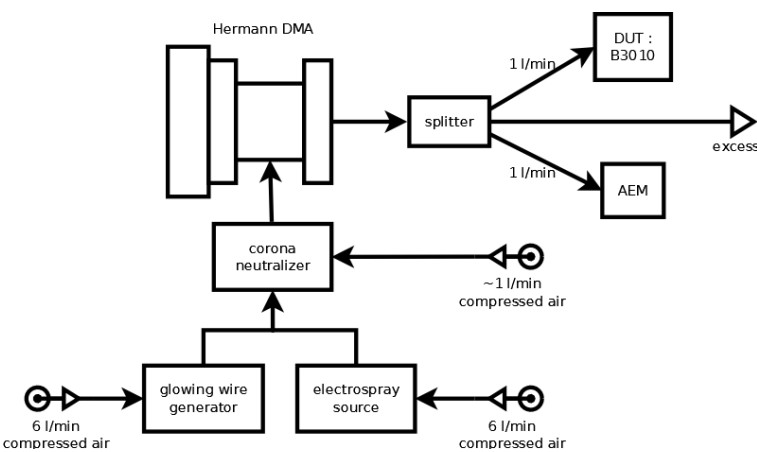

**Figure 2.** Experimental setup

The experimental setup given in the Fig. 2 is the classical setup and method used during the last decades for the generation of
sub-10 nm and detection efficiency measurements with a high particle size resolution. It has been widely documented in the literature (Heim et al. (2010), Jiang et al. (2011), Kangasluoma et al. (2015), Hering et al. (2016), Kangasluoma et al. (2017), Barmpounis et al. (2018)).

The differential mobility analyzer (DMA) used in this study is called a Hermann-type DMA and has been described in details in Kangasluoma et al. (2016). DMAs are operated with 2 flows : $Q_a$, the aerosol (or sample) flow, and $Q_s$, a filtered, aerosol-
free sheath flow. The size resolution of a DMA is given by the ratio $Q_a/Q_s$. Typical DMAs are operated at $Q_a$ = 1-4 l/min and $Q_s$ = 5-20 l/min. The Hermann-type DMA used in this study is operated at $Q_a$ = 10 l/min and $Q_s$ = 250-1500 l/min. The much higher $Q_a/Q_s$ ratio is the key parameter to select aerosol particles with high resolution. A high resolution DMA is needed because the particles used to measure the detection efficiency are in a very narrow size range. With Fig. 3, we can calculate the resolution of the DMA, defined by the full width at half maximum of the peak (FWHM) over the central size of the peak
(Kangasluoma et al. (2016)). The FWHM of 0.1 nm over the monomere size 1.47 nm gives a resolution of 0.07 (dimensionless number) in the conditions of the experiments.

However, this high resolution comes at a price : the small particle size range. Indeed, at such a high sheath flow rate, the voltage required to select particles bigger than 5-6 nm produces electric arcs in the DMA, and thus sets the upper limit. Besides,

the principle of linking the DMA voltage and actual particle size is based on the DMA voltage at which the peaks of a molecular standard of known size are resolved (Ude and Fernandez De La Mora (2005)). The relationship between voltage and size is established in the conditions of the experiment and remains valid provided the conditions do not change. Reducing the sheath flow rate would allow to select bigger particles, but their size would be unknown.

Two different types of aerosols are used in this study to test the response of the B3010 in the sub-3 nm range. Mobility standard ions generated with an electrospray source for organics, and metal oxides produced with a glowing wire generator for hydrophobic particles, are used sequentially as the sources of polydisperse aerosols in front of the high resolution Hermann type DMA (Kangasluoma et al. (2016)). The DMA is run in a closed loop arrangement. Both methods produce self, singly charged aerosol particles.

Nitrogen is used as a carrier gas for the wire generator and the electrospray source to push the particles into the Hermann DMA at a flow rate of 6 l/min. Indeed, the transmission in the DMA is higher in for sub-3 nm particles when the aerosol flow rate is higher than 5 l/min (Kangasluoma et al. (2016)).

The monodisperse aerosol flow exiting the DMA is distributed with a 3-port flow splitter to the device under test (B3010), the Keithley 6517B reference aerosol electrometer (AEM) and an exhaust line for the excess air. Conductive soft tubes of equal lengths are used to connect the splitter to the B3010 and the AEM, in order to level off the deposition losses in both lines.

The flow rate in the loop of the DMA is constant but is not measured nor known. The flow control uses the actual speed output of the high flow blower to control the flow rate. Tetraheptylammonium bromide is used as a standard to calibrate the flow rate of the DMA at the beginning of the experiments, and to check the stability of the system afterwards. Then, a different type of aerosol can be injected into the DMA, because the parameters of the DMA don't change as long as the flow rates are kept constant. The calibration factor k, determined by calibration is needed to relate the particle mobility Z to the measured voltage V. The factor k is then given by the expression :

$$k = \frac{Z \cdot V}{Q} = Z \cdot V = cst \tag{3}$$

k is constant if Q is constant. Q is the total flow rate in the DMA, which in our case can be approximated to the sheath flow rate. Q is assumed to be constant hereafter. Z is the mobility diameter and V the voltage in the DMA. The mobility diameter $Z_s$ of the monomer THA$^+$ is selected by the DMA for a voltage $V_s$. $Z_s$ = 1/1.03 cm$^2$ V$^{-1}$ s$^{-1}$ is given by Ude and Fernandez De La Mora (2005).

As k is a constant, we can write :

$$k = V_s \cdot Z_s = V \cdot Z \tag{4}$$

Where V is the voltage required to select particles of mobility Z with the DMA.

Eq.4 is then used to transform the horizontal axis of the measured distribution concentration versus voltage given by the scanning ramp of the inner electrode of the DMA to the distribution concentration mobility.

The mobility diameter of the measured mobility distribution is then converted to mobility diameter using the Stockes-Cunningham equation (Friedlander (2000)) :

$$
\begin{cases}
Z_{S_c} = n \cdot e \cdot \dfrac{1 + K_n \cdot (A + B \cdot e^{\frac{-C}{K_n}})}{3 \cdot p \cdot \mu \cdot d_z} & (5) \\[4mm]
K_n = 2\dfrac{\lambda_g}{d_z} \qquad\qquad\qquad \lambda_g = \dfrac{\mu}{\rho_g}\sqrt{\dfrac{\pi \cdot M_g}{2 \cdot R \cdot T}} & (6)
\end{cases}
$$

Where $d_z$ (noted $d$ in the rest of the paper) is the mobility diameter, $\mu$ is the dynamic viscosity, n is the number of elementary charges e born by the ion or particle, $K_n$ is the Knudsen number, $\lambda_g$ is the mean free path of the molecules of the carrier gas, $\rho_g$ its density and T its temperature. The constants A, B and C are taken from Friedlander (2000).

## 3.2 Molecular standards

We use the molecular standards listed in Table 1 to generate particles in the nanometer range with the electrospray source. The standards are dissolved in ethanol, at a concentration of 1 mmol/l, according to the method described in Ude and Fernandez De La Mora (2005). The dissolved compounds can qualify as standards because 1/ they produce individual molecules whose size is stable and known, and 2/ because the DMA resolution is high enough to separate the peaks of the individual molecules from each other.

| Abbr. | Formula | Name | Soluble in water |
|-------|---------|------|:----------------:|
| TBAB | $C_{16}H_{36}BrN$ | Tetra-n-butylammonium bromide | no |
| TXAB | $C_{24}H_{52}BrN$ | Tetrahexylammonium bromide | yes |
| THAB | $C_{28}H_{60}BrN$ | Tetraheptylammonium bromide | no |

Table 1. Molecular standards dissolved in ethanol

The monomer and dimer of the molecular standards are so small, that they can only bear a single electric charge. But most particles larger than about 1.8 nm carry multiple charges, and hence generate a much higher current in the electrometer than if they all had borne a single charge. So, the electrometer overestimates the particle number concentration by at least one order of magnitude. As a result, the CPC detection efficiency is underestimated for these particles.

Traditionally, the molecular standards have mostly been used up to about 2 nm. But in order to measure the ability of the B3010 to detect molecular standard ions at sizes larger than 2 nm, To do so, we insert a small corona discharge device, operated at a DC voltage of about 3 kV. The sign of the voltage is the opposite of that of the charge of the particles selected in the DMA. The benefit of the corona discharge can be seen in Fig. 3.

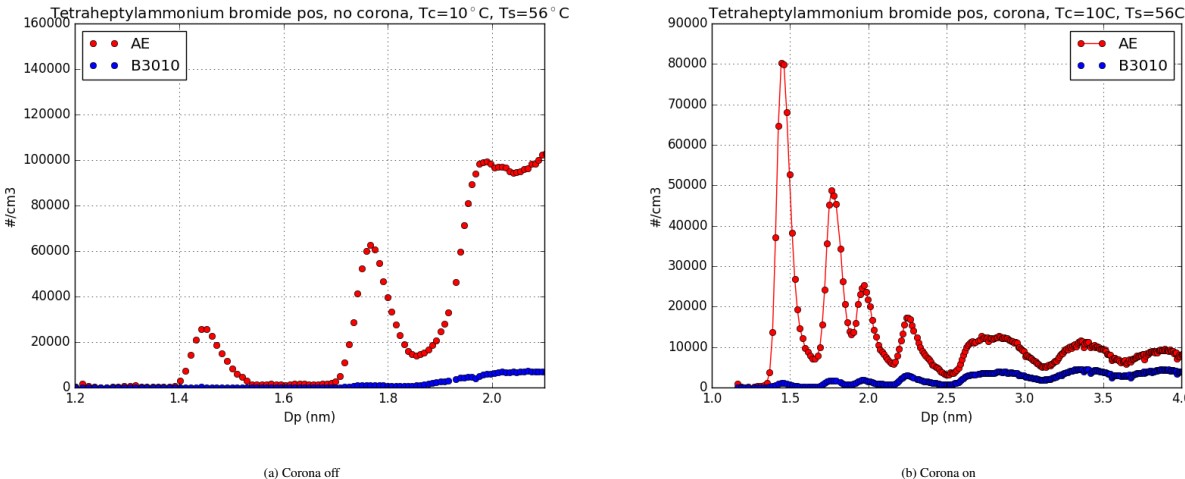

**Figure 3.** Effect of the corona discharge. When the corona discharge is off (a), the signal of the aerosol electrometer (AE) rises sharply above 2 nm. When the corona discharge is on (b), the DMA can resolve up to the 4th peak.

### 3.3 Glowing wire generator

In the second set of experiments, we use hydrophobic oxide particles produced by wire a glowing wire generator, as described in Peineke et al. (2006) and Kangasluoma et al. (2013). The metal wire is heated resistively with a DC current of several Amps.

When it reaches a high enough temperature, the wire starts to glow, like the filament of a light bulb. It is installed in a stainless steel ISO KF NW40-flanged cross : gas inlet faces gas outlet, and a view port faces the wire mount. The wire is flushed with a 10 l/min nitrogen stream to carry the produced material towards the DMA and to keep the temperature of the wire well bellow the melting point. The wire is the hottest point of the generator. The volatile material condenses by homogeneous nucleation as soon as it drifts from the hot wire surface to form self-charged, positive and negative particles, from a few nanometers to a

few tens of nanometers.

In this study, we use a tungsten oxide alloy (WOX) ⌀1.0×100 mm wire.

## 4 Experimental results and discussion

### 4.1 Laboratory calibration

The detection efficiency is the most representative characteristic of a CPC, and is what we focus on in this section. The detection

efficiency $\eta$ of the CPC is defined as the ratio between the particle concentration measured by the CPC, $N_{CPC}$ to the particle concentration given by the aerosol electrometer, $N_{AE}$ for different diameters, signs or $\Delta T$, according to Eq. 7.

$$\eta = \frac{N_{CPC}}{N_{AE}} \tag{7}$$

We measured the detection efficiency of the B3010 for three molecular standards, including positively and negatively charged particles for a ΔT of 46 °C and Tcond of 10°C (Fig. 4). The sign of the charge of the particles depends on the sign of the voltage applied to the electrospray generator. The B3010 exhibits the same cut-off diameter of about 3 nm for all three standards, soluble or not, demonstrating the relative insensitivity of butanol to the chemical composition of particles (Kangasluoma et al. (2014)).

However, we can see in Fig. 4 that the detection efficiency is higher for negative particles, compared to positive particles. This phenomenon is known as *sign preference*, and has been observed by a number of studies before us. More than a century ago, Wilson (1897, 1899) reported that more fog is formed in an expansion chamber when negative ions are present compared to positive ions. Also, more fog forms in the presence of bipolar ions compared to no ions at all. Wilson is cited by McMurry (2000). More recent studies (Winkler et al. (2008), Kangasluoma et al. (2013)) reported the same thing. This observation is thus in accordance with previous studies.

The efficiency should only increase with increasing particle diameter. However, the curves in Fig. 4 reach a plateau. Despite the corona discharge, the efficiency is clearly underestimated for particles bigger than about 3 nm to 4 nm, for the reasons detailed earlier.

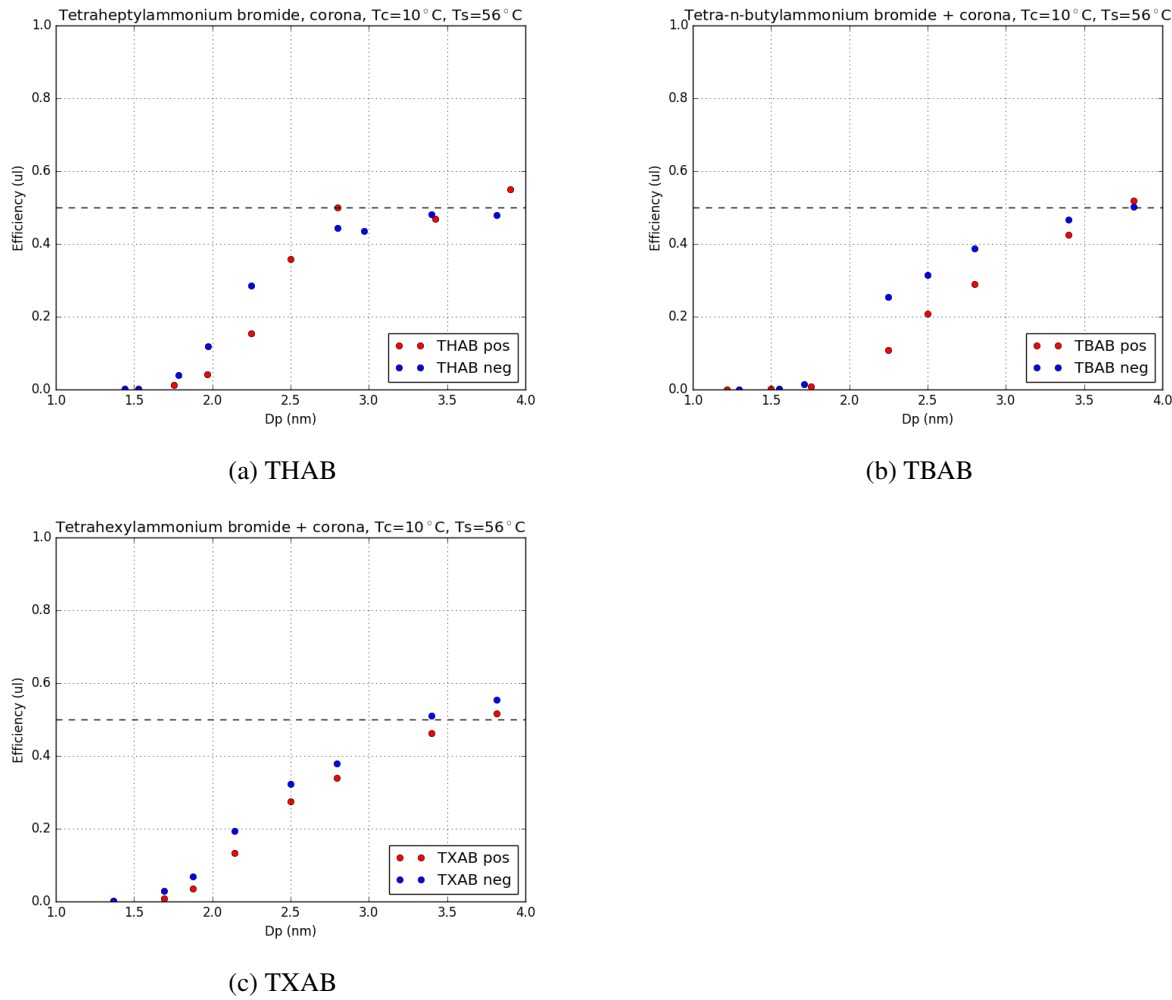

(a) THAB

(b) TBAB

(c) TXAB

**Figure 4.** Detection efficiency curves measured for various laboratory-generated aerosols.

Fig. 5 illustrates the dependence of the B3010 detection efficiency on the sample flow rate, for two different ΔT. We observe that the efficiency does not vary monotonically with respect to the flow rate. As reported by Kuang et al. (2012) and Kangaslu-oma et al. (2015), this is the evidence of a competition between diffusion losses and the time spent in the supersaturated flow. At low flow rates, the diffusion losses are higher, but the particles have more time to activate and grow to a detectable size. At higher flow rates, the diffusion losses are smaller, but the residence time is too short for the droplets to grow to a detectable size. Indeed, when the flow rate increases, the saturation profile develops further down the condenser tubes, thus contributing to decrease the residence time of the particles in supersaturated conditions. The detection efficiency peaks at 1 l/min, which is exactly the nominal flow rate of the TSI 3010. We can notice here that increasing ΔT from 17 °C to 47-48 °C does not shift the optimum flow rate value.

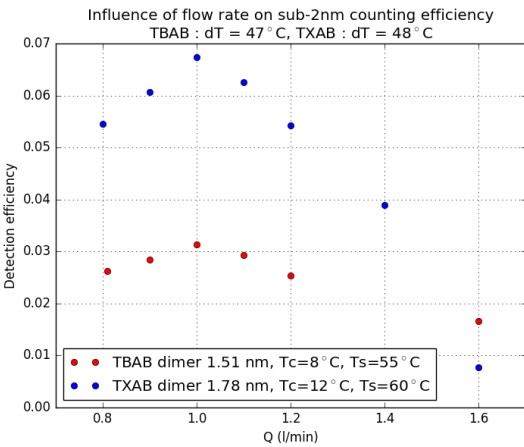

**Figure 5.** B3010 detection efficiency as a function of flow rate in the sub-2 nm range.

In order to study the effect of pressure on the detection efficiency, we inserted a pinched tube section in the setup of Fig. 2, between the DMA and the flow splitter. We favor a pinched tube over a needle valve since it features lower particle losses. By reducing this variable tube section, we are able to reduce the pressure in both the aerosol electrometer and the B3010. A pinched tube has a greater penetration efficiency than a needle valve. The effect of the inlet pressure on the B3010 detection efficiency is illustrated in Fig. 6.

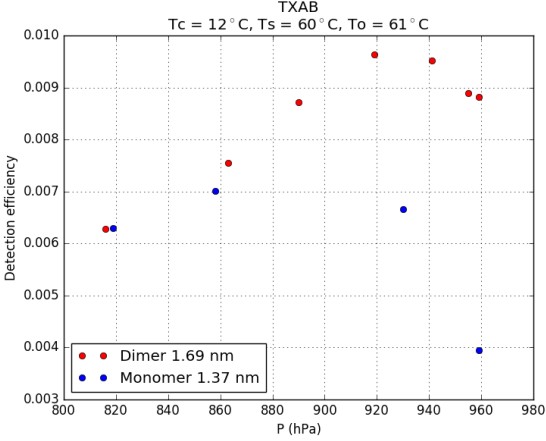

**Figure 6.** B3010 detection efficiency as a function of inlet pressure for two fixed-sized standards : TXAB monomer and dimer.

As the pressure decreases in the condenser, the mean free path of the particles increases as well. The probability for them to hit the walls is greater. This leads to an increase in diffusion losses, especially for the smaller, more mobile particles.

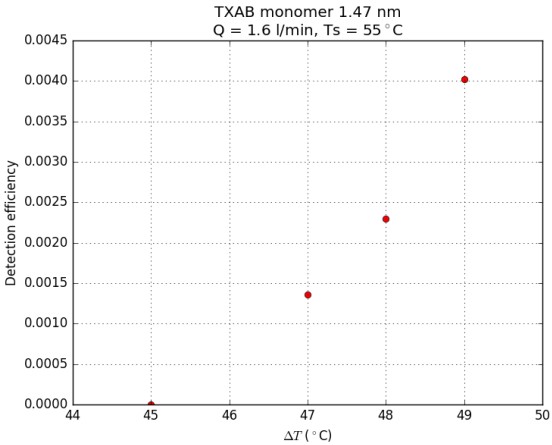

**Figure 7.** B3010 detection efficiency as a function of temperature gradient for TXAB monomer.

Fig. 8 shows how the detection efficiency increases when the temperature gradient $\Delta T = T_s - T_c$ increases. For a $\Delta T$ of 40°C, the cut-off diameter is 3 nm, while for a $\Delta T$ of 45°C, the cut-off diameter is 2.5 nm. Homogeneous nucleation was never observed under our experimental conditions. We also observe, as already reported by Barmpounis et al. (2018), that not only the amplitude of the temperature gradient impacts the detection efficiency, but also the shift of this "temperature window" in the temperature domain. The B3010 thus has the same detection efficiency if $\Delta T = 48$°C at $T_c = 12$°C or if $\Delta T = 47$°C at $T_c = 8$°C. The efficiency curves of two ultra-fine CPCs, taken from the product specification sheets, are plotted alongside the B3010 calibration data on Fig. 8 for comparison.

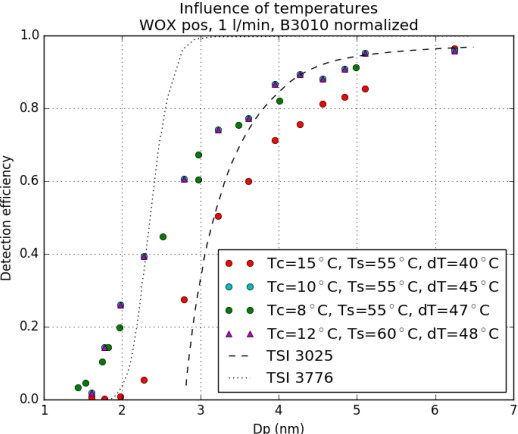

**Figure 8.** Influence of temperatures on detection efficiency of tungsten oxide particles. The black lines are the typical curves of ultrafine CPCs TSI 3025 ($D_{P50}$ = 3.0 nm) and TSI 3776 ($D_{P50}$ = 2.5 nm).

The increase in performance achieved in this development is shown in Fig. 9. The B3010, a "boosted" version of the TSI 3010 competes almost with ultra-fine CPCs. Again, as the flow path was not optimized for sub-10 nm particles, the curve is not as steep as those of ultra-fine CPCs.

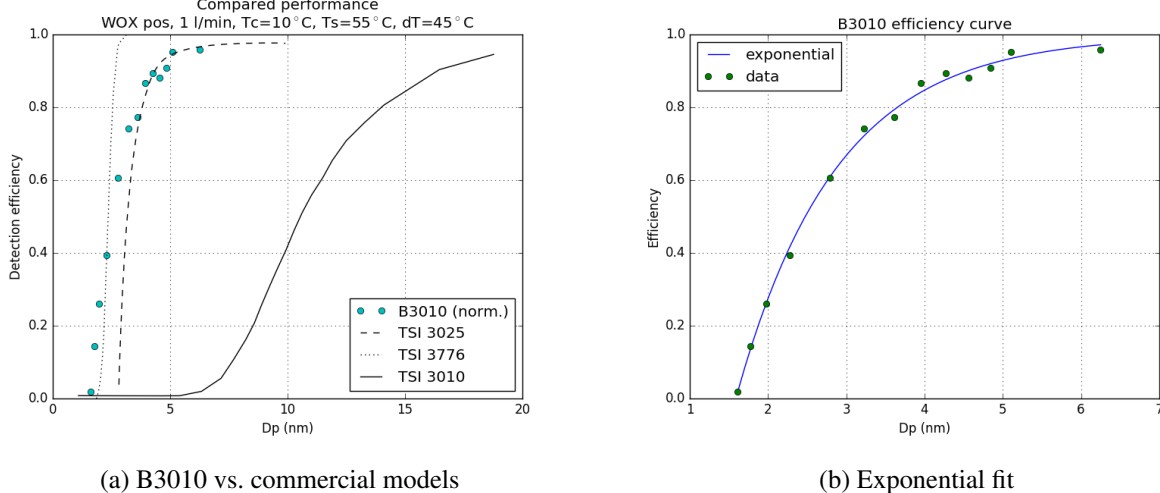

(a) B3010 vs. commercial models       (b) Exponential fit

**Figure 9.** Gain in performance. Tungsten oxide particles. The data from TSI CPCs are taken from product brochures (a). The B3010 efficiency is fit with an exponential curve (b) to retrieve the cut-off diameter. The data from the B3010 are normalized.

5      The cut-off diameter is defined as the particle size at 50 % efficiency, and is abbreviated $D_{P50}$. We applied the exponential fit of Eq. 8 to the data in Fig. 9.

$$\eta = y_0 - e^{\frac{x_0 - x}{k}} \tag{8}$$

Where $\eta$ is the detection efficiency, x is the particle size, and $y_0$, $x_0$ and k are the coefficients of the fit. This type of fit is commonly used in the community (Wiedensohler et al. (1997)). We can calculate $D_{P50}$ by evaluating the inverse fit function 10    for a detection efficiency $\eta = 0.5$. The associated error is given by Eq. 9, where R is the resolution of the DMA calculated in section 3.1. The 1/2 factor comes from the fact that R is calculated from FWHM. We can thus tell that $D_{P50} = 2.5 \pm 0.1$ nm.

$$\epsilon = \frac{D_{P50} \cdot R}{2} \tag{9}$$

The response time of our CPC was measured in Enroth et al. (2018). It was found that the B3010 has a response time similar to that of the TSI 3010, *i.e.* about 2.3 s. This is not surprising, since both models share the same geometry.

## 4.2 Ambient measurements

Finally, we had the B3010 measure ambient air in a suburban location close to Clermont-Ferrand (France), alongside a TSI 3025 and a TSI 3010 for three days. The settings were $T_c = 10°C$, $T_s = 56°C$ and Q=1.0 l/min. The B3010 is corrected for coincidence with Eq. 1.

The maximum particle counting rate is limited by the coincidence in the optics. The higher the number of particles flowing through the optics, the higher the probability of coincidence. In the TSI 3025, only a fraction of the intake air is sampled, the rest being used to make filtered sheath air. Thus, this dilution reduces the number of particles flowing through the optics, and hence the coincidence phenomenon. According to the manual, it can count up to $10^5$ #/cm$^3$ without coincidence. On the other hand, in the TSI 3010, the sample air is not diluted, and the maximum measurable concentration without coincidence is $10^4$ #/cm$^3$. As the B3010 has the same design and optics as the TSI 3010, we expect a similar concentration range. Fig. 10a focuses on a *high concentration* episode, when the concentration peaks above $10^4$ #/cm$^3$. The B3010 curve crosses the TSI 3025's at about $1.7×10^4$ #/cm$^3$. Fig. 10b shows a *low concentration* episode, when the concentration was below $10^4$ #/cm$^3$ at all times. In these conditions where the comparison is fairer, the concentration of the B3010 is significantly higher than that of the TSI 3025, which is in accordance with Fig. 9a.

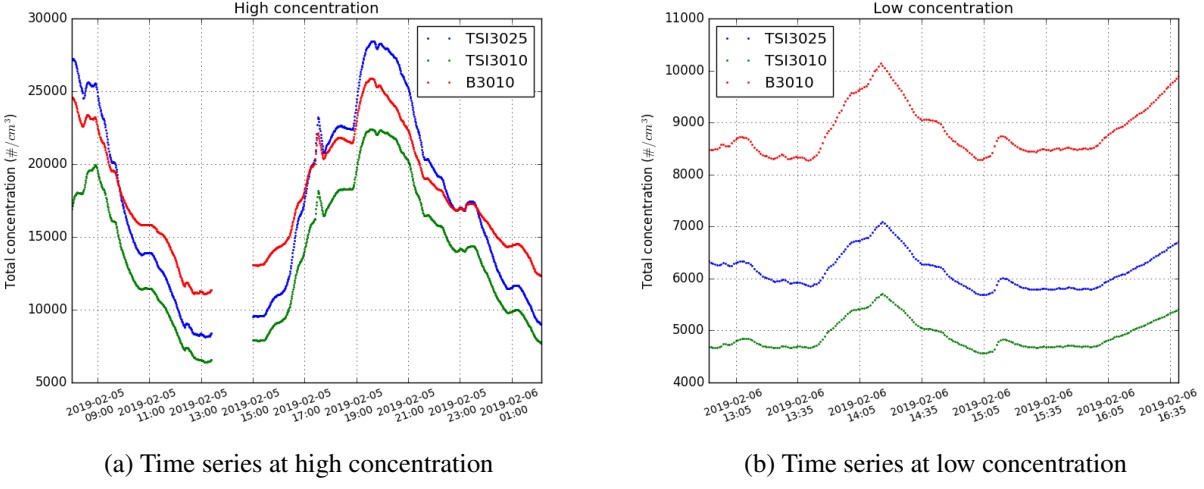

(a) Time series at high concentration          (b) Time series at low concentration

**Figure 10.** Comparing B3010 with TSI 3025 and TSI 3010 during a high (a) and a low concentration episode (b).

In Fig. 11, we show an example application of the B3010 to retrieve the size distribution at the lower particle diameters, like in Kangasluoma et al. (2014). We plot three particle size bins, obtained by the difference of the concentrations N reported by CPCs pairs :

- small : B3010 minus TSI 3025, 2.5 < N < 3.0 nm

- medium : TSI 3025 minus TSI 3010, 3.0 < N < 10 nm

– large : TSI 3010, N > 10 nm

In this example, one can see that particles in the range 2.5-3.0 nm account for about a third of the total particle concentration.

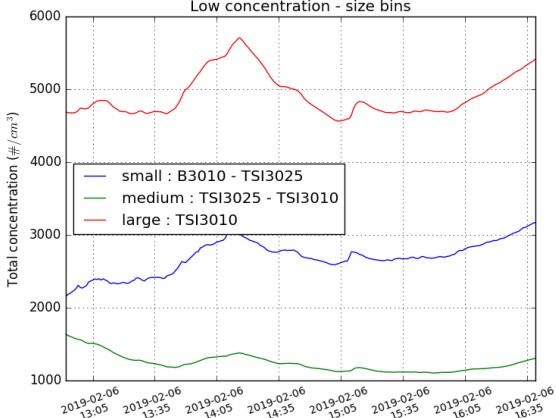

**Figure 11.** The B3010 in a CPC battery.

## 5   Conclusions

In this project, we demonstrate that a CPC with a simple, proven design such as the TSI 3010's can be slightly modified to bring the detection efficiency close to that of ultra-fine, sheathed CPCs. Both laboratory and ambient measurements confirm this result. However, as emphasized in the ambient measurements section, the maximum measurable concentration is definitely limited by the design of the flow path and of the optics.

   In the light of the results presented here, a few guidelines for CPC designers and users can be drawn. Still today, designers

could reasonably imagine non-sheathed, yet performing CPC geometries, as long as the total concentration in the targeted application is not too high. Besides, owners of non-sheathed CPCs can dramatically reduce the cut-off diameter of their devices, by simply adjusting the saturator and condenser temperatures. This simple tweak, when allowed by commercial CPC firmwares, can potentially help save the extra cost of an ultra-fine CPC.

*Author contributions.*   David Picard wrote the first draft and ran the ambient measurements. Michel Attoui provided the laboratory calibration

facility. David Picard, Michel Attoui and Karine Sellegri participated in laboratory tests, data analysis and redaction of the paper.

*Competing interests.*   No conflicting interests are present between any of the authors and any company cited in the paper.

*Acknowledgements.* This work was funded by the ClerVolc project - Programme 1 "Detection and characterization of volcanic plumes and ash clouds" funded by the French government "Laboratory of Excellence" initiative.

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
