# Peer review of "B3010: A Boosted TSI 3010 CPC for Airborne Studies"

_Atmospheric Measurement Techniques, 2018_

## Referee Comment (RC1) · Anonymous Referee #3 · 27 Dec 2018

The paper describes a method to boost the performance of the condensational particle counter CPC TSI 3010 used originally in monitoring of clean rooms, but also in atmospheric applications, e.g. in airborne measurements. The instrument is already not produced, however it is widely used and available on second-hand market, thus the method is of boosted performance in atmospheric (particularly airborne) applications is possible interest to many researchers from e.g. cloud physics community and/or aerosol community. The proposed boost is aimed at enhancement of detection limit in small aerosol particles from 1 nm in the original design down to 1.5 nm in the boosted version, labeled B3010.

The basic physical principle on which the improvement in the performance nab be obtained is change of the operation temperatures of the counter.

[Figure]

In the boosted version the saturator, condenser and the optics of the original 3010 model are used, however improvements in the temperatures of saturator and condenser as well as in the volume flow rate are introduced. The electronics and control elements of the system is projected and introduced from scratch.

The new B3010 was tested in the laboratory using two substantially different types of sub 3 nm particles: electrospray meeting molecular standards (for organics) and metal oxides from glowing wire generator (for non-organic).

The tests indicate noticeable detection efficiency of both, organic and non-organic particles beginning from ∼2nm comparably to the existing ultrafine CPCs TSI 3015 and TSI 3776.

Criticism

While the idea of enhancing the performance of the existing CPC is appealing and important, the main drawback of the work id lacking direct laboratory comparison to the existing instruments of similar detection range. I do understand that this wad behind the reach of the authors, however to convince the readers to the performance of the boosted sensor more detailed description and discussion is needed.

Correction suggestions: 1) Figure 1. The scheme of B3010 should indicate which elements have been improved/changed compared to the standard 3010 instrument. The adequate edits tn the modification description should be added in the text.

2) Expand, please the section 3, provide more details on the DMA (Dynamic mechanical Analyzer) and the reference AEM used.

3) Results and discussion section should be expended and better explained. Why do you see remarkable differences in detection for positively and negatively charged particles? How do you measure pressure? Figures in this section need better descriptions, some error analysis should be added into the text. Also more discussion real measurements and comparison between B3010 and 3025 should be more detailed. Do you

observe drift in the performance of the sensor in the course of the tests, which Fig. 10 suggest? If so why?

---

## Author Comment (AC1) · 11 Mar 2019

article

**General comments**

We, the authors, would like to thank Anonymous Reviewer # 3 for his/her interest in our work, and for suggesting changes for the sake of clarity and quality of the paper.

**Specific comments**

**1. Figure 1**

*Reviewer : "Figure 1. The scheme of B3010 should indicate which elements*

[Figure]

*have been improved/changed compared to the standard 3010 instrument. The adequate edits tn the modification description should be added in the text."*

We reworked figure 1 (instrument diagram) to show the B3010 and the TSI 3010 side by side. It will look much like the figure attached to this reply.

In the paper, the text right bellow Fig. 1 lists the changes we made from the original TSI 3010 model. In a nutshell, it mainly consists in :

- thermally decoupling the condenser from the saturator to allow independent temperature control

- controlling the flow rate (when the internal pump is connected in place of the critical orifice).

The saturator, the condenser and the optics are those from the original TSI 3010.

**2. DMA and electrometer**

***Reviewer : "Provide more details on the DMA (Dynamic mechanical Analyzer) and the reference AEM used"***

In the scope of this paper, DMA stands for *"Differential Mobility Analyzer"*. The DMA will be better described in the next version of the paper, section "3.1 Experimental setup". We plan to add the following text :

*"The DMA used in this study is called a Hermann type DMA and has been described in details in Kangasluoma et al. 2016. DMAs are operated with 2 flows : $Q_a$, the aerosol (or sample) flow, and $Q_s$, a filtered, aerosol-free sheath flow. The size resolution of a DMA is given by the ratio $\frac{Q_a}{Q_s}$.*

*Typical DMAs are operated at $Q_a$ of 1-4 l/min and $Q_s$ of 5-20 l/min. The Hermann type DMA used in this study is operated at $Q_a$ = 10 l/min and $Q_s$ in the range of 250-*

*1500 l/min. The much higher $\frac{Q_a}{Q_s}$ ratio is the key to select aerosol particles with high resolution.*

*A high resolution DMA is needed because the particles used to measure the detection efficiency are in a very narrow size range. With Fig. 3b, we can calculate the resolution of the DMA, defined by the full width at half maximum of the peak (FWHM) over the central size of the peak (Kangasluoma et al. 2016). The FWHM of 0.1 nm over the monomere size 1.47 nm gives a resolution of 0.07 in the conditions of the experiments."*

The electrometer used in this study is a Keithley 6517B. This will be mentioned in the paper.

**3. Results and discussion**

***Reviewer : "Why do you see remarkable differences in detection for positively and negatively charged particles?"***

This is a well-known phenomenon known as *"sign preference"*. We would like to add the following text to the paper, in section "4.1 Laboratory calibration" :

*"We can see in Fig. 4 that the counting efficiency is higher for negative particles, compared to positive particles. This phenomenon is known as "sign preference", and has been observed by a number of studies before us. Already a century ago, Wilson 1897; 1899 reported that more fog is formed in an expansion chamber when negative ions are present compared to the presence of positive ions, and also more fog formation in the presence of bipolar ions compared to no ions at all. Wilson is cited by McMurry 2000. More recent studies (Winkler et al. 2008, Kangasluoma et al. 2013) reported the same thing. Our work is thus in accordance with previous studies."*

These additional references will be added in the next version of the paper :

- Wilson, C. T. R. (1897). Condensation of Water Vapour in the Presence of Dust-

Free Air and other Gases. Philosophical Transactions of the Royal Society of London 189:265-307.

- Wilson, C. T. R. (1899). On the condensation nuclei produced in gases by the action of röntgen rays, uranium rays, ultra-violet light, and other agents. Philosophical Transactions of the Royal Society of London 192:403-453.

- McMurry, Peter H. (2000). The History of Condensation Nucleus Counters,Aerosol Science and Technology, 33:4, 297-322

- Winkler, Paul M. (2008). Heterogeneous Nucleation Experiments Bridging the Scale from Molecular Ion Clusters to Nanoparticles. Science 07 Mar 2008, Vol. 319, Issue 5868, pp. 1374-1377, DOI: 10.1126/science.1149034

***Reviewer : "How do you measure pressure?"***

The absolute pressure sensor is a FirstSensor HDI0611ARY8P3 (600-1100 mbar). The differential pressure sensor is a FirstSensor LDES050UE3S (50 Pa, unidirectional). The flow rate was calibrated with a Drycal Gilibrator bubble volume flow meter for a number of absolute pressures.

The section "2. Design of the B3010" of the paper will contain more details about pressure and flow measurement, with the addition of the following paragraph : *"The absolute pressure is measured by a miniature sensor, connected to the optical chamber with a capillary tube. The pressure intake is centrally located, between the saturator-condenser block and the laminar flow element. Besides, given the low pressure drop in the flow path, the absolute pressure does not change much from the inlet to the laminar flow element. The volume flow rate is calculated from the differential pressure measured by a 50 Pa miniature sensor accross a laminar flow element and compensated for absolute pressure. The volume flow rate is a key measurement, since it is used to calculate the particle number concentration. The concentration $C$ is calculated*

from the number of particle counts $N$, accumulated during $T_S$, at a volume flow rate $Q_V : C = \frac{N}{Q_V \cdot T_S}$"

***"Reviewer : "Figures in this section need better descriptions, some error analysis should be added into the text."***

Most figures in this section show the detection efficiency of the device under test, with respect to the reference aerosol electrometer.

The detection efficiency E of the CPC is given by the ratio between the particle concentration measured by the CPC, $C_{CPC}$ to the particle concentration given by the aerosol electrometer, $C_{AE}$ for different diameters, signs or $\Delta$T :

$$E = \frac{C_{CPC}}{C_{AE}}.$$

To calculate the diameter at 50 % efficiency, $D_{P50}$, we applied an exponential fit to the data from Fig. 9 of the paper :

$$E = y_0 - e^{\frac{x_0-x}{k}}$$

Where $x$ is the particle size, and $y_0$, $x0$ and $k$ are the coefficients of the fit. This type of fit is commonly used by Global Atmospheric Watch aerosol calibration centers (*e.g.* TROPOS, Leipzig, Germany).

Evaluating the inverse fit function at a counting efficiency $E = 0.5$, and knowing the resolution of the DMA, we can tell that $D_{P50} = 2.5 \pm 0.1 \, nm$ (see Fig. 1 attached).

We plan to add this to the paper.

***Reviewer : "Also more discussion real measurements and comparison between B3010 and 3025 should be more detailed. Do you observe drift in the performance of the sensor in the course of the tests, which Fig. 10 suggest? If so why?"***

Indeed, in the first submitted version, the measurements used to make Fig. 10 were not correct. A closer look at the data actually showed a drift of the B3010. It was probably caused by a too high room temperature (higher that 30 degC), that must have caused the condenser temperature of the B3010 to stray from the setpoint, thus reducing the counting efficiency.

As a consequence, we ran another ambient air test at room temperature of 20 degC. In addition to the TSI 3025, we also compare the B3010 to a TSI 3010, sampling every second, and using 1-minute averages for the plots. The total counting efficiency of the B3010 vs. that of the TSI 3025 is now in accordance with the rest of the article. In Fig. 8, for example, the efficiency of the B3010 is higher than that of the TSI 3025. This is also the case in this ambient test, when concentrations bellow $10,000 \ \#/cm^3$ are considered.

In the figures attached to this reply, we call "low concentration" conditions the time period when the concentration was bellow $10,000 \ \#/cm^3$. In Fig. 2 attached (3 CPCs time series), we can see that the B3010 displays a higher concentration than the TSI 3025, which is itself higher than the TSI 3010. We plot in Fig. 3 attached three particle size bins, obtained by the difference of the concentrations reported by CPCs pairs :

- small : B3010 minus TSI 3025, 2.5-3.0 nm

- medium : TSI 3025 minus TSI 3010, 3-10 nm

- large : TSI 3010, > 10 nm

One can see that particles in the range 2.5-3.0 nm account for about a third of the total particle concentration.

We will update the ambient air experiment in the next version of the paper.

---

## Author Comment (AC2) · 11 Mar 2019

See figures hereafter.
* * *
[Figure]

**Fig. 1.** B3010 and TSI3010 diagrams

**B3010 efficiency curve**

Efficiency vs Dp (nm)

- exponential
- data

**Fig. 2.** B3010 counting efficiency fit

[Figure]

[Figure]

**Fig. 3.** B3010 vs. TSI3025 and TSI3010 (time series)
**Low concentration - size bins**

Total concentration ($\#/cm^3$)

- small : B3010 - TSI3025
- medium : TSI3025 - TSI3010
- large : TSI3010

**Fig. 4.** B3010 vs. TSI3025 and TSI3010 (size bins)

---

## Referee Comment (RC2) · Anonymous Referee #4 · 14 Mar 2019

This generally well-written paper describes a method to push the minimum detectable particle size down to 1.5 nm. The authors refer to this as "boosting" the performance; a term of which I am not so fond. The formerly available CPC TSI 3010 has been modified and is appropriate for aircraft use for reasons discussed in the text. The instrument would also remain useful in ambient studies. The instrument has been out of production for a number of years, and I am skeptical of the claims "widely available on the second-hand market"; from where? I cannot see any on eBay, for example. I consider revising this statement, but agree that there are many currently in use around the world.

Though modifying and enhancing the performance of old instrumentation is a worthwhile pursuit, it would have been extremely useful to have compared this instrument

directly across all main measurement metrics, with TSI's 3756, which measures down to 2.5nm. Figure 9 would have been very interesting with such a comparison.

Further to the other reviewer's comments (which I agree with), Fig.6 detection efficiency curves are not really pleasing at all. Why not increase the particle diameter to ensure that a DE of 1 is obtained?

Figure 10b shows a systematic error to my eye; the B3010 is always undercounting, not just above 1E4/cc. I would like to see this plot on a log-log axis, and again - does the B3010 agree with a 3025 for say PSL at 300nm? There we should be getting 1:1 agreement to within 5% at least. Commercial CPCs can agree to within 1-2%, and though it's useful to push the detection limit as low as possible, if the detection efficiency at larger diameters is not 100% then there's limited use for a modified CPC.

I think you should comment, or take measurements of, the rise-time. If you're using such a CPC in aircraft, the response time is important (arguably it is in any platform). To that end, mixing-type CPCs have faster time responses than saturator and laminar flow CPCs, and should the airmass change, how long for the B3010 to react to this? I believe this data should be presented (such as changes to a filtered air mass).

There should be a comparison of different particle types of the same "size". e.g. explicit response to silver particles, sodium chloride, PSL, etc. . . and not at the lower diameter. 100nm would be fine for this. Does the CPC respond the same way as a normal CPC? Figure 4 shows that the CPC never detects all particles, but. . . surely it must do at large enough diameters? I would like to see figure 4 expanded to 500nm for example.

Basically, there should be a more rigorous analysis of this CPC compared with commercially available CPCs, namely the 3756 as this detects down to 2.5nm, but it's unlike the authors have access to this, so a 3775/6 would also suffice.

Also, I am not sure the main advantage of "lower cost" is true, and really the main advantages of such an undertaking should be for the scientific merit. I think the conclu-

sions need expansion and are generally lacking. You should report how well the B3010 agrees with other CPCs under "normal" conditions, and also how it compares at the smaller end. What is meant by "outlook of future CPCs should be simplicity"?

[Figure]

[Figure]

(a) THAB

(b) TBAB

[Figure]

(c) TXAB

**Figure 4.** Efficiency curves measured for various laboratory-generated aerosols.

**Fig. 1.**

---

## Author Comment (AC3) · 20 Mar 2019

article

**General comments**

We, the authors, would like to thank Anonymous Reviewer # 4 for his/her interest in our work, and for making interesting suggestions.

**Specific comments**

*Reviewer : "I am skeptical of the claims 'Widely available on the second-hand market' ; from where ?"*

[Figure]

Some companies specialize in used instrument retail, *e.g.* in the USA. Instruments of the TSI 3010 family are still available today.

***Reviewer : "it would have been extremely useful to have compared this instrument directly across all main measurement metrics, with TSI's 3756, which measures down to 2.5nm. Figure 9 would have been very interesting with such a comparison."***

The counting efficiency of the TSI 3776 is plotted in Fig. 9. Its cut-off diameter is 2.5 nm as well.

***Reviewer : "Figure 10b shows a systematic error to my eye; the B3010 is always undercounting, not just above 1E4/cc. I would like to see this plot on a log-log axis, and again - does the B3010 agree with a 3025 for say PSL at 300nm? There we should be getting 1:1 agreement to within 5% at least. Commercial CPCs can agree to within 1-2%, and though it's useful to push the detection limit as low as possible, if the detection efficiency at larger diameters is not 100% then there's limited use for a modified CPC"***

We would like to draw the attention of reviewer #4 to the reply to RC1 : Fig. 10 will be updated with a new data set. The new time series figure shows that the total counting efficiency of the B3010, at concentrations lower than $10,000 \ \#/cm^3$, is much higher than that of the TSI 3025, whose cut-off diameter is 3 nm. This, and Fig. 9, show that the efficiency does rise up to 1, with increasing particle diameters.

***Reviewer : "I think you should comment, or take measurements of, the rise-time."***

Our B3010 took part in Enroth et al. 2017, where it was found that the B3010 has a response time similar to that of the TSI 3010, *i.e.* about 2.3 s. This is not surprising, since both models share the same geometry.

*Reviewer : "Figure 4 shows that the CPC never detects all particles, but. . . surely it must do at large enough diameters? I would like to see figure 4 expanded to 500nm for example."*

As said in the paper (p9, lines 1-4), the molecular standards bear a lot of multiple charges that increase the current in the electrometer, hence the low plateau.

As detailed in reply to RC1, the Herrmann-type DMA has a very high resolution of about 0.1 nm at smaller particle diameters. This performance is obtained by a very high sheath flow rate of about 1000 l/min. At such a high sheath flow rate, the voltage required to select particles bigger than 5-6 nm produces electric arcs in the DMA, and thus sets the upper limit. Besides, the principle of linking the DMA voltage and actual particle size prevents to change the sheath flow rate during the experiment.

As a consequence, it is absolutely impossible to extend this plot up to 500 nm.

*Reviewer : "I think the conclusions need expansion and are generally lacking."*

We will rework the conclusion in the final version.

Our message is that designers can imagine non-sheathed, yet performing CPC geometries. Our work could also help users select a model to purchase, and get the most out of their non-sheathed CPC with a simple tweak.

---

## Author Response (AR1)

**General comments**

We, the authors, would like to thank the reviewers for their interest in our work, and for suggesting changes for the sake of clarity and quality of the paper.

In the replies to Referee Comments (RC1 and RC2), we cited a number of extra references. We added these references to the new version of the paper.

We homogenized the terms "counting efficiency" and "detection efficiency" to only use the latter.

In this document, deleted text appears in red and inserted text in blue.

**Specific comments**

**1 Reviewer # 3**

**0 1.1 Figure 1**

15

Reviewer: "Figure 1. The scheme of B3010 should indicate which elements have been improved/changed compared to the standard 3010 instrument. The adequate edits to the modification description should be added in the text."

We complemented Fig. 1 (instrument diagram) to show the B3010 and the TSI 3010 side by side and updated the legend accordingly. This was a very wise suggestion, since the new figure lets the reader capture our modifications in the blink of an eye.

The text does describe these modifications further bellow. They consist mainly in 1/ thermally decoupling the condenser from the saturator and 2/ controlling the flow rate.

The saturator, the condenser and the optics are those from the original TSI 3010.

The new Fig. 1 is now:

Figure 1. B3010 (left) and TSI 3010 (right) diagrams

**1.2 DMA and electrometer**

**Reviewer: "Provide more details on the DMA (Dynamic mechanical Analyzer) and the reference AEM used"**

In the scope of this paper, DMA stands for "Differential Mobility Analyzer". The DMA is now better described in section "3.1 Experimental setup", and specifically how the flow rates in the DMA affect the size resolution. In addition, we calculate the size resolution of the DMA.

**We added:**

The differential mobility analyzer (DMA) used in this study is called a Hermann-type DMA and has been described in details in Kangasluoma et al. (2016). DMAs are operated with 2 flows:  $Q_a$ , the aerosol (or sample) flow, and  $Q_s$ , a filtered, aerosol-free sheath flow. The size resolution of a DMA is given by the ratio  $\frac{Q_a}{Q_s}$ . Typical DMAs are operated at  $Q_a = 1$  to 4 l/min and  $Q_s = 5$  to 20 l/min. The Hermann-type DMA used in this study is operated at  $Q_a = 10$  l/min and  $Q_s = 250$ -1500 l/min. The much higher  $\frac{Q_a}{Q_s}$  ratio is the key parameter to select aerosol particles with high resolution. A high resolution DMA is needed because the particles used to measure the detection efficiency are in a very narrow size range. With Fig. ??, we can calculate the resolution of the DMA, defined by the full width at half maximum of the peak (FWHM) over the central size of the peak (Kangasluoma et al. (2016)). The FWHM of 0.1 nm over the monomere size 1.47 nm gives a resolution of 0.07 in the conditions of the experiments.

The electrometer used in this study is a Keithley 6517B.

We replaced the reference aerosol electrometer by the Keithley 6517B reference aerosol electrometer.

**1.3 Results and discussion**

**Reviewer: "Why do you see remarkable differences in detection for positively and negatively charged particles?"**

This is a well-known phenomenon known as *"sign preference"*, first observed by Wilson in the late 19th century. In section "4.1 Laboratory calibration", we give a list of references to papers that report the same phenomenon.

**We added:**

However, we can see in Fig. ?? that the detection efficiency is higher for negative particles, compared to positive particles. This phenomenon is known as *sign preference*, and has been observed by a number of studies before us. More than a century ago, Wilson (1897, 1899) reported that more fog is formed in an expansion chamber when negative ions are present compared to positive ions. Also, more fog forms in the presence of bipolar ions compared to no ions at all. Wilson is cited by McMurry (2000). More recent studies (Winkler et al. (2008), Kangasluoma et al. (2013)) reported the same thing. This observation is thus in accordance with previous studies.

**Reviewer: "How do you measure pressure?"**

- The section "2. Design of the B3010" of the paper now contain more details about absolute pressure and flow measurements. The absolute pressure is measure in the optical chamber. The volume flow rate is measured with a laminar flow element and a differential pressure sensor as shown in Fig. 1. The absolute pressure sensor is a FirstSensor HDI0611ARY8P3 (600-1100 mbar). The differential pressure sensor is a FirstSensor LDES050UE3S (50 Pa, unidirectional). The flow rate was calibrated with a Drycal Gilibrator bubble volume flow meter for a number of absolute pressures.
- In addition to these technical details, we explain how important the flow rate measurement is, since it is used to calculate particle number concentration.

**We added:**

The absolute pressure is measured by a miniature sensor, connected to the optical chamber with a capillary tube. The pressure intake is centrally located, between the saturator-condenser block and the laminar flow element. The volume flow rate is calculated from the differential pressure measured by a 50 Pa miniature sensor across a laminar flow element and compensated for absolute pressure. The volume flow rate was calibrated with a Drycal Gilibrator bubble volume flow meter for a number of absolute pressures. The volume flow rate is a key measurement, since it is used to calculate the particle number concentration.

The concentration C is calculated from the number of particle counts N, accumulated during  $T_S$ , at a volume flow rate  $Q_V$ , according to Eq. 2.

$$C = \frac{N}{Q_V \cdot T_S} \tag{2}$$

5 "Reviewer: "Figures in this section need better descriptions, some error analysis should be added into the text."

In section "4.1 Laboratory calibration", we give the definition of the detection efficiency of a CPC: the ratio of the concentration given by the CPC over that given by the reference electrometer.

We added:

The detection efficiency is the most representative characteristic of a CPC, and is what we focus on in this section. The detection efficiency  $\eta$  of the CPC is defined as the ratio between the particle concentration measured by the CPC,  $C_{CPC}$  to the particle concentration given by the aerosol electrometer,  $C_{AE}$  for different diameters, signs or  $\Delta$ T, according to Eq. 7.

$$\eta = \frac{C_{CPC}}{C_{AE}} \tag{7}$$

Further down in the same section, we updated Fig. 9 with a fit plot to calculate the diameter at 50 % efficiency,  $D_{P50}$ . And knowing the resolution of the DMA calculated earlier, we can calculate the error associated to the value of  $D_{P50}$ . We added a curve fit in Fig. 9b, and the text bellow it:

**Figure 9.** Gain in performance. Tungsten oxide particles. The data from TSI CPCs are taken from product brochures (a). The B3010 efficiency is fit with an exponential curve (b) to retrieve the cut-off diameter. The data from the *B3010* are normalized.

The cut-off diameter is defined as the particle size at 50 % efficiency, and is abbreviated  $D_{P50}$ . We applied the exponential fit of Eq. 8 to the data in Fig. 9.

$$\eta = y_0 - e^{\frac{x_0 - x}{k}} \tag{8}$$

Where  $\eta$  is the detection efficiency, x is the particle size, and  $y_0$ ,  $x_0$  and k are the coefficients of the fit. This type of fit is commonly used in the community (?). We can calculate  $D_{P50}$  by evaluating the inverse fit function for a detection efficiency

 $\eta = 0.5$ . The associated error is given by Eq. 9, where R is the resolution of the DMA calculated in section ??. The 1/2 factor comes from the fact that R is calculated from FWHM. We can thus tell that  $D_{P50} = 2.5 \pm 0.1 \ nm$ .

$$\epsilon = \frac{D_{P50} \cdot R}{2} \tag{9}$$

The response time of our CPC was measured in Enroth et al. (2018). It was found that the B3010 has a response time similar to that of the TSI 3010, *i.e.* about 2.3 s. This is not surprising, since both models share the same geometry.

Reviewer: "Also more discussion real measurements and comparison between B3010 and 3025 should be more detailed. Do you observe drift in the performance of the sensor in the course of the tests, which Fig. 10 suggest? If so why?"

Indeed, in the first submitted version, the measurements used to make Fig. 10 were not correct. A closer look at the data actually showed a drift of the B3010 probably caused by a too high room temperature (above 30 degC).

As a consequence, we ran another ambient air test at room temperature of 20 degC. In addition to the TSI 3025, we also compare the B3010 to a TSI 3010, for 3 days. The total counting efficiency of the B3010 vs. that of the TSI 3025 is now in accordance with the rest of the article. In Fig. 8, for example, the efficiency of the B3010 is higher than that of the TSI 3025. This is also the case in this ambient test, when concentrations bellow  $10,000 \#/cm^3$  are considered.

In the updated Fig. 10, we distinguish 2 domains of high and low concentration, the limit between them being  $10,000 \#/cm^3$ . We removed:

Fig. 10b actually shows that B3010 starts to drift significantly from the 1:1 line at concentrations greater than  $10^4 \#/cm^3$ . We added:

Fig. 10a focuses on a *high concentration* episode, when the concentration peaks above  $10^4 \#/cm^3$ . The *B3010* curve crosses the TSI 3025's at about  $1.7 \times 10^4 \#/cm^3$ . Fig. 10b shows a *low concentration* episode, when the concentration was below  $10^4 \#/cm^3$  at all times. In these conditions where the comparison is fairer, the concentration of the *B3010* is significantly higher than that of the TSI 3025, which is in accordance with Fig. 9a.

Figure 10. Comparing B3010 with TSI 3025 and TSI 3010 during a high (a) and a low concentration episode (b).

In the new Fig. 11, we show a possible use case for the B3010, as part of a CPC battery. We added:

In Fig. 11, we show an example application of the *B3010* to retrieve the size distribution at the lower particle diameters, like in Kangasluoma et al. (2014). We plot three particle size bins, obtained by the difference of the concentrations reported by CPCs pairs:

- small: B3010 minus TSI 3025, 2.5-3.0 nm

- medium: TSI 3025 minus TSI 3010, 3-10 nm

- large: TSI 3010, > 10 nm

In this example, one can see that particles in the range 2.5-3.0 nm account for about a third of the total particle concentration.

Figure 11. The B3010 in a CPC battery.

**5 **2 Reviewer # 4**

**2.1 Introduction**

Reviewer: "I am skeptical of the claims 'Widely available on the second-hand market'; from where ?"

Some companies specialize in used instrument retail, e.g. Artisan TG, USA. Instruments of the TSI 3010 family are still available today.

**10 We removed:**

its wide availability on the second-hand market.

We added:

Besides, it is still pretty easy to buy instruments from the TSI 3010 family from companies that specialize in used scientific instruments retail.

**15 2.2 Results and discussion**

Reviewer: "it would have been extremely useful to have compared this instrument directly across all main measurement metrics, with TSI's 3756, which measures down to 2.5nm. Figure 9 would have been very interesting with such a comparison."

The counting efficiency of the TSI 3776 is plotted in Fig. 9. Its cut-off diameter is 2.5 nm, just like TSI 3756.

20 Reviewer: "Figure 10b shows a systematic error to my eye; the B3010 is always undercounting, not just above 1E4/cc. I would like to see this plot on a log-log axis, and again - does the B3010 agree with a 3025 for say PSL at 300nm? There we

should be getting 1:1 agreement to within 5% at least. Commercial CPCs can agree to within 1-2%, and though it's useful to push the detection limit as low as possible, if the detection efficiency at larger diameters is not 100% then there's limited use for a modified CPC''

The new ambient test time series (Fig. 10) shows that the total counting efficiency of the B3010, at concentrations lower than  $10,000 \ \#/cm^3$ , is much higher than that of the TSI 3025, whose cut-off diameter is 3 nm. This, and Fig. 9, show that the detection efficiency does rise up to 1, with increasing particle diameters.

Please refer to section 1.3 of this document, especially to the updated Fig. 10 and the new Fig. 11.

**Reviewer: "I think you should comment, or take measurements of, the rise-time."**

Our B3010 took part in a laboratory study, where it was found that the B3010 has a response time similar to that of the TSI 3010, *i.e.* about 2.3 s. This is not surprising, since both models share the same geometry.

We added:

The response time of our CPC was measured in Enroth et al. (2018). It was found that the B3010 has a response time similar to that of the TSI 3010, *i.e.* about 2.3 s. This is not surprising, since both models share the same geometry.

Reviewer: "Figure 4 shows that the CPC never detects all particles, but... surely it must do at large enough diameters? I would like to see figure 4 expanded to 500nm for example."

As said in the original submission version (p9, lines 1-4), the molecular standards bear a lot of multiple charges that increase the current in the electrometer artificially, hence the low plateau of the detection efficiency.

In order to justify why we can't extend Fig. 4 up to 500 nm, we now give more details about the Herrmann-type DMA in section "3.1 Experimental setup". The DMA has a very high resolution of about 0.1 nm, but the biggest selectable particle is only about 5-6 nm. This is due to the relationship between the sheath flow rate and the DMA voltage. In a nutshell, we trade off size range, because we cannot compromise resolution.

We added:

However, this high resolution comes at a price: the small particle size range. Indeed, at such a high sheath flow rate, the voltage required to select particles bigger than 5-6 nm produces electric arcs in the DMA, and thus sets the upper limit. Besides, the principle of linking the DMA voltage and actual particle size is based on the DMA voltage at which the peaks of a molecular standard of known size are resolved (?). The relationship between voltage and size is established in the conditions of the experiment and remains valid provided the conditions do not change. Reducing the sheath flow rate would allow to select bigger particles, but their size would be unknown.

**2.3 Conclusion**

**Reviewer: "I think the conclusions need expansion and are generally lacking."**

We rewrote the conclusion in the new version.

Our message is that designers can imagine non-sheathed, yet performing CPC geometries. Our work could also help users select a model to purchase, and get the most out of their non-sheathed CPC with a simple tweak.

We removed

[revised manuscript text omitted]